# The Wellbeing of Chinese Migrating Grandparents Supporting Adult Children: Negotiating in Home-Making Practices

**DOI:** 10.3390/ijerph19169903

**Published:** 2022-08-11

**Authors:** Dan Zhu, Haichao Xu, Yuan Yao

**Affiliations:** 1Glorious Sun School of Business & Management, Donghua University, Shanghai 200051, China; 2Tourism College, Hunan Normal University, Changsha 410081, China

**Keywords:** wellbeing, home, migrating grandparents, home-making practice, Chinese context, qualitative study

## Abstract

Health geography provides a relational approach to understanding elders’ wellbeing experience in relation to place. That the migrating grandparents move between their home and their adult children’s home to support their children’s life in the migrating city provides a particular pattern to supplement the place-based wellbeing literature. How they negotiate their wellbeing remains to be observed in the daily home-making practices related to their two homes. This study conducted in-depth interviews with 35 migrating grandparents and nine of their adult children and conducted extensive field notes in Shanghai from 2020 to 2022. Through thematic analysis, it finds that the migrating grandparents met a series of differences, challenges and tensions in the material, social and emotional home-making practices brought by the separation and rotation between their own and their children’s homes. It weakens their physical, social and mental wellbeing. However, they take some initiatives to overcome and relieve these tensions. Therefore, accompanied by sacrifices and negotiations, they also obtain sustained material, social and spiritual–emotional values to negotiate a suboptimal experience of wellbeing. This study contributes to the intersection of elderly wellbeing and home-making studies by revealing the complex and ongoing inter-relationships between migrating grandparents and home in the rotating lifestyle.

## 1. Introduction

The wellbeing of the elderly has received multi-disciplinary research attention, including economy, psychology, sociology and geography [1]. Among them, geographers pay special attention to the role of place that constantly shapes elders’ wellbeing experiences [2,3,4,5,6,7]. Conventional studies explore the unidirectional effects of place on elders’ wellbeing by individually considering the contextual and compositional elements of place [4,8]. In this sense, a place is treated with physical boundaries at a specific scale. Home, at both the household and the community scales, is focused as a relatively stable space of security, familiarity and autonomy where the elderly live, which is crucial for their physical, social and mental wellbeing [9,10,11,12,13,14,15].

The relational approach that emerged in health geography in the last 20 years challenges the conventional view of place [2]. It treats place connecting different nodes (people, physical settings, material goods, artifacts, etc.) as relational networks at multi-scales. Place is not separated by physical distance, but is (re)territorialized and de-territorialized by social–relational distance, hence is relatively dynamic and fluid. Moreover, people’s daily and life-long course mobilities are emphasized as an important dimension influencing the social–relational distance [2]. Individuals’ practice can be seen as reciprocal human–place interactions during which people can obtain subjective feelings about how they live [4]. Such a relational approach proposes two challenges to the elderly home-based wellbeing studies. First, wellbeing is not a stable or integrated mindset but an ongoing experience through continuous negotiations with the place. Second, home is not a physical container with fixed boundaries, but is a fluid and dynamic network in which elders’ daily practices at multi-scales constantly construct its meaning. 

Elders’ periodic mobility among multiple places further complicates how elders negotiate their wellbeing with multiple homes. Previous studies have shed light on the amenity-led seasonal retirees’ wellbeing experience in moving between their original and second homes [4,16,17]. In comparison, the seniors who are busy moving between their own and their adult children’s homes to support their children’s families are not carefully examined [18,19,20,21]. Rotating between the two homes becomes a lifestyle for these grandparents. Sometimes, they leave their hometown and live with their children and grandchildren for a long time to undertake the home-caring responsibilities. Intermittently they go back to their own home and live their own life.

Such a moving lifestyle shows two particularities to advance the migrating elders’ home-based wellbeing research. First, it differs from the pure labor migrants who move for work nor the amenity-led consuming migrants. These grandparents are not only consumers of the migrating city but also labor forces free of charge to their children’s families [22]. Second, as they move between their own home and the children’s home, they are simultaneously related to the two homes in their negotiations of wellbeing experience. When they live in the migrating home, they may adapt to the differences from those in their own home and cooperate with their offspring to conduct the home-making practices. This is different from the regular migrants who are entirely autonomous with the routine life in their new home space [17,23,24,25]. When they come back to their own home, their social and spiritual connections with the migrating home where their children live may also arise. Hence, how these migrating elders proceed with their wellbeing within this moving lifestyle is firmly connected with how they balance their relations with the two homes in daily practices. 

Therefore, it remains to dig up the home-making practices of these migrating grandparents to see how they adapt to the shared home life, overcome the separation of the two homes and achieve physical, social and mental wellbeing. Currently, it is relatively neglected in the literature. Hence, this paper aims to explore the dynamic wellbeing experience of these migrating grandparents by focusing on the elder-home interrelations in their home-making practices. To this end, we take Chinese migrating grandparents moving to Shanghai, the biggest city in China, as a typical case. 

The paper is structured as such. We combed through the literature on both the well-being of elders and the practice of home-making. Through the relational approach, we identified the research gap on how the migrating grandparents negotiate their wellbeing in their home-making practices. This is followed by a review of the Chinese socio-cultural context of migrating grandparents. After the method section, we reported our findings about the migrating grandparents’ wellbeing experience in their balancing with the relations to their separated homes in three aspects: material, social, and emotional. The discussion and conclusion section further highlights the theoretical and practical contributions. 

## 2. Literature Review

### 2.1. The Wellbeing of Elderly and Home: A Relational Approach

Wellbeing is an important concern for aging seniors and has attracted multi-disciplinary research interests. A consensus is that both the hedonic (a utilitarian view emphasizing satisfaction of preferences) and the eudaimonic (emphasizing personal growth and self-realization) approaches complement each other in conceptualizing wellbeing [1,26]. Schwanen and Atkinson categorized wellbeing studies into two general directions. One is the “components approach to wellbeing”, the other “the relational and processual conceptions of wellbeing” [7] (p. 99). The former explores wellbeing as measurable and governable components which people can acquire as desirable states of being [7]. The latter develops from the conventional place-based wellbeing studies and perceives wellbeing as dynamic effects through human–place relations among the material, environment, discourse and bodily practices [2,5,27,28]. In this process, wellbeing experience can be conceived through three dynamically-evolved dimensions: physical, social and mental wellbeing [4]. Physical wellbeing refers to the physiological health of the human body. Social wellbeing pertains to a good state of social contacts and support. Mental wellbeing refers to people’s positive feelings, emotions and spiritual attachments [29]. Table 1 summarizes the differences between conventional and relational approaches to understanding place and wellbeing. 

Previous studies have highlighted the crucial role of the home in influencing elders’ wellbeing experience [14,28,33,34]. The relational approach helps to reconsider the home-based wellbeing studies for elders in three points. First, elders cannot be treated as stable groups belonging to a particular location. Through daily practices related to different scales (e.g., having breakfast on the household scale; taking exercises on the community scale; phoning children in another city on the regional scale, etc.), they are mobile in diverse temporal–spatial patterns. This brings the second point that home is not a stable entity or a physical container with fixed geographical boundaries. Through elders’ diverse daily practices, it is fluidly assembled by the human, material, social–cultural elements at multi-scales. Elders’ practice plays as the “constellations of connections” that forms the reciprocal interrelationship between the elderly and the home [2,35] (p. 178). Third, elders’ wellbeing experience may be dynamically negotiated through their home-related daily practices in the material, social and emotional aspects. In the material aspect, previous studies have shown that arranging the material environment, the housing and the financial conditions of the home afford the basic conditions for aging people’s inhabitation where they maintain their physical health and develop a psychological sense of security [33,36]. In the social–cultural aspect, the stable acquisition of social relations with relatives, friends and neighbors always fosters elders’ sense of home and elevates their self-efficacy, life satisfaction and place attachment [14,36]. In the mental and emotional aspect, building emotional and spiritual attachments with other family members or the aged caring institutions in the community may significantly fulfill the elders’ mental wellbeing [36,37,38]. 

For those elders who step into a moving lifestyle, home seems to be more unstable and separated for them. Thus, a more challenging and complicated picture to piece together these temporary and multi-site homes is formed. The elderly may have to adapt to the new home’s resources and materials through their corporeal body and bear the loss of the familiar social ties in their original home [39]. They may also face the challenge of losing a sense of home, unsettling their emotions on the move [40,41]. Under this circumstance, what kind of elder–home relationship would be constructed and influence the seniors’ wellbeing? This makes us turn to the current literature about home-making practices on the move with a focus on the older migrants.

### 2.2. Home-Making and the Older Migrants 

Home-making has increasingly attracted academic interest in the past 20 years, mainly in international mobilities [20,25,42,43]. A core research thread is how migrating people keep the balance between life on the move and attaching a sense of stability and belonging to the mobile places. Home as simultaneously sedentary and nomadic should be reconfigured [43]. Hence home-making is studied as a process of people’s continual practices in daily life to eliminate this tension brought by mobilities [25,43]. Currently, empirical research about home-making mainly centers on transnational migrants [24,25,44], transnational laborers [45] and nomadic homeless people [46]. Their home-making practices include the materiality, social relations and emotions connected with migrants’ negotiation of belongings, residence with mobile places and landscapes, and how they develop them [47]. 

Specifically for the elderly, previous home-making studies have paid some attention to the family task-oriented migrating grandparents, who move from their hometown to the big metropolis domestically or transnationally to support their kids’ families [20]. They become the vital force for these overworked modern families [48,49]. Earlier literature mainly focuses on the migrating grandparents’ care arrangements, roles and responsibilities, social adaptation or integration into the new country or city, and the intergenerational relations within the migrating home [20,21,50,51,52,53]. For example, Ducu and Nedelcu classify the care arrangements carried out by European and Non-European migrant grandparents in Switzerland into family support in childbirth circumstances, troubleshooting in occasional need of childcare, full and permanent childcare and family support, and intergenerational sharing and transmission [20]. 

Along with the relational turn, it gradually emphasizes a de-territorialization of home in home-making studies [54]. Home is not fixed to a single location. As Nowicka pointed out, “home should be seen as something that individuals can take along as they move through time and space” [55]. This requires a broader perspective on the migrating elders’ home-making practices, simultaneously considering their dynamic relations to both the original and the migrating homes in material, social and emotional aspects during the moving process and their wellbeing experience becomes a layered process composed through these embodied home-making practices [56,57]. 

Such a relational perspective becomes more prominent to consider the grandparents migrating domestically. Previous studies mainly focus on the transnational migrating grandparents who deal with long-haul distance, low moving frequency and significant cultural differences [58]. To the domestic migrating grandparents, the differences between the originating and destination places may be smaller and they can migrate more frequently and smoothly between homes. Such a flexible pattern of mobility may influence their home-making practices in relation to the two separated homes. It is more interesting to explore how these migrating grandparents maintain and coordinate their wellbeing experience in this process.

Moreover, previous literature about migrating grandparents is mainly grounded in European and American societies. More empirical studies in different contexts are warranted. This study makes up for these gaps by investigating the domestic migrating grandparents rooted in China. Hence the Chinese context is reviewed as follows. 

### 2.3. Migrating Grandparents in the Chinese Context

Based on the Confucian tradition, Chinese people put great emphasis on family and home [59]. The culture of the Chinese home is firmly connected with kinship and roots. With the bond of blood kinship, family members live together for generations without leaving the homeland [60]. Hence, home is a crucial place affecting the wellbeing of the Chinese elderly [61]. It can be reflected in Chinese slang, such as “cheng huan xi xia” (elders’ happiness comes from living together with their children and being cared for by them) and “san dai tong tang” (three generations of grandparents, parents and grandchildren living together, indicating harmony and a happy life). Traditionally, home is a place with stable, permanent, and harmonious geographical imaginations.

With rapid Chinese urbanization, especially in the past 40 years, many young talents flowed from rural and suburban areas to big cities for education and employment. Some of them settle down and organize their families there. However, as they are new citizens, these young couples have to struggle hard to afford the high housing price, especially in the first and second-tier cities in China, and also to obtain “Hukou”, a permanent certificate of citizenship. To deal with such great living pressure, most of them become dual-earner families. Since the governmental power has withdrawn from the public childcare field, these newly-born families have to spend a lot on employing housework and caring workforce from the market [62]. To eliminate the burden on these adult children, a large number of grandparents leave their own homes to support the new homes of their children and constantly move between them. The traditional stable home connecting blood kinship and native land, therefore, breaks up. 

Then, through the relational approach, how do the migrating grandparents negotiate their relations with the two separated homes through home-making practices? How do they proceed with their wellbeing in this process? These are questions that this paper aims to answer (Figure 1).

## 3. Method

### 3.1. The Case City

We took Shanghai as the case immigrant city for this study. It is a prominent city with the largest population and economic volume in China, attracting many migrating workers and talents each year. The latest census conducted in Shanghai 2021 shows that the migrant population in Shanghai is 10.48 million. Its proportion increased from 25.4% in 2010 to 42.1% in 2021 [63]. Therefore, many new migrating families form in this city. Hence, Shanghai is a typical city for us to research the migrating grandparents supporting their adult children.

### 3.2. Data Collection

The qualitative approach is chosen as this study aims to explore the migrating grandparents’ wellbeing experiences by looking into their home-making practices. A semi-structured in-depth interview is mainly conducted, which has been widely used in previous place-based wellbeing and home-making studies [4,17,20,64]. The interview is conducted in two stages. The first and third authors conducted the first stage of interview from 15 January to 20 February 2020, mainly through WeChat and phone calls. Through acquaintances’ introduction and snowball sampling, we conducted 30 interviews with migrating grandparents aged 51 to 75. They mainly come from the neighboring provinces of Shanghai. To make data triangulation, we simultaneously interviewed nine adult children of some of these grandparents [65]. This helped collect more details about how grandparents are involved in the daily home-making practices concerning their two homes. The second interview stage was conducted in May 2022 by the first and second authors through face-to-face or WeChat voice calls when we supplemented five in-depth interviews of migrating grandparents, also through acquaintances’ introduction.

In between the two stages of interview, the first author frequently conducted informal face-to-face talks with her parents, who are also migrating grandparents, and the other unfamiliar migrating grandparents in the community. Informal talk is commonly used in qualitative studies to understand the phenomenon and supplement the formal data collection [66]. These informal talks greatly informed the authors to better understand the home life and the inner thoughts of these migrating grandparents, including their household work details, their social activities within the communities, and their contacts and comments about the moving lifestyle and their two homes. The first author kept writing field notes about them. Based on these, the interview at the second stage only asked the questions that presented conflicting answers between the previous formal interviews and the informal talks. These questions were fully discussed with each of the five interviewees. Hence large samples were not pursued at this stage. Table 2 and Table 3 show the profiles of all interviewees.

All three authors have systematically learned the qualitative study method. The first author has over ten years of experience in qualitative study, and she supervised the whole process. The interview questions surrounded the material, social and emotional aspects of home-making practices and their connections with the grandparents’ physical, social and mental experiences of wellbeing. Typical questions are: How do you arrange the daily home issues in the immigrant city? How do you perceive and feel about the moving lifestyle? How do you perceive the two homes you rotate and their influence on your physical and mental health, capabilities, self-efficacy, and spiritual belongings? The interview time ranges from 15 to 40 min. It ranges along with the interviewees’ different willingness to disclose their inner thoughts. 

### 3.3. Data Analysis

After each round of data collection, we transcribed all the interview recordings into texts and read the transcripts and field notes to fully understand the home-making stories and personal feelings. Thematic analysis was adopted manually through thematic charts as it is suitable to extract the themes about the negotiation of wellbeing in home-making practices and is frequently used in the literature on elders’ place-based aging and wellbeing [67,68]. Based on previous literature, we first divided the data into three categories, material, social, and emotional/spiritual home-making practices (including the corresponding narration of wellbeing experience). In each category, we classified the data into three sub-categories by distinguishing the elders’ narration of their physical, social and mental wellbeing experiences. Table 4 shows some operational labels of these categories/sub-categories.

We coded the data iteratively and inductively [4]. To ensure reliability and validity, peer check, debriefings and discussions were frequently conducted among the three authors during the coding process [73,74]. After frequent comparison with the literature to improve transferability [73], core themes emerged about how the migrating grandparents negotiate their wellbeing experience during the rotating lifestyle. To improve the credibility of the analysis, we use some representative quotations of the data in the report of our research findings [73]. 

## 4. Findings

### 4.1. Material Practices: Sacrifices and Gains

Once moving to Shanghai, the grandparents dedicated their efforts to the material practices in their offspring’s home. Most interviewees reported their adaptation to various differences from those in their own homes, during which they experienced physical and mental sacrifices, adjustments and gains. 

The sacrifice is typically reflected in their concession in the usage of home space and the labor dedication of their corporeal bodies. For three generations living together, allocating the home space is a big challenge. Our research finds that most migrating grandparents self-consciously leave more home space for their children and grandchildren. Just as A05 describes,


*“I should leave some space for the children. It’s not easy for them (after working) a whole day. Let them three watch TV and do something of their own in the living room…I just browsed the mobile phone in my bedroom. Now the mobile phone is very convenient.”*
(A05)

A05 represents those whose adult children get home from work on time and do not need to look after their grandchildren in the evening. However, many other migrating grandparents have to care for the kids all day long. A11 reports his schedule one day: 


*“After breakfast, we send our grandson to school. Then we have housework to do. We wash clothes and clean the house. It takes almost the whole morning. Then we read the news on the phone. Sometimes we go out for a walk. After lunch in the afternoon, we have a rest for an hour. Around 2 o’clock, we go out for a walk. Then we get our grandson back. I accompanied my grandson to study. Sometimes I asked him to write and read. After supper, I took him out to play in the small park. He played with other children. At 7 o’clock, we would come back to watch TV. At 9 o’clock, I would wash my face and go to sleep. That’s the end of my day.”*
(A11)

A11 reflects the hard physical strength paid by these migrating grandparents. In this sense, except for the living room and their bedroom, the other rooms seem purely the production space as the elders have to do cleaning, washing, cooking or baby-caring work the whole day. To match the work intensity of their adult children, they have to squeeze time to rest. A05 also use the word “fight” and “busy” to describe one day.

Therefore, it dramatically challenges the physical wellbeing of the migrating grandparents. Most of them do not report having a big illness but some common chronic diseases (e.g., hypertension) that can be controlled through daily medication. Their physical health is quite important to the migrating family. Thus, their children also keep a close eye on it.


*“I usually pay more attention to my mother’s health. After all, she*
*’s old. I tell her if feeling uncomfortable, speak out and don’t hold on*
*… body comes first.”*
(B08)


*“If the elders get sick, the family will be in disorder.”*
(B06)

Hence, their adult children’s care gives these grandparents some psychological comfort. All the interviewees expressed that they would insist on supporting their children as long as their body permits. They always buy medicines wholesale and apply for medical reimbursement when returning to their home city to save money for their children.

Moreover, the grandparents also gain a better economic state as most of their daily expenditure is included in the living expenses paid by their adult children. 


*“We basically bear all (the daily expenses). It’s already so hard for them (their parents) to come to help us. Their round-trip fare, we also pay.”*
(B05)

With the financial support of their children, the migrating grandparents arrange the everyday recipes for the family. However, as they come from different parts of China, many of them are not adapted to the ingredients and tastes in Shanghai. Some from the remote areas even could not adapt to the climate and environment of Shanghai and experienced bodily discomfort.


*“I’m not used to the food in Shanghai. The food back home tastes salty and spicy, but the food in Shanghai tastes very sweet.”*
(A29)


*“At the beginning, when I came to Shanghai, I was a little uncomfortable. Three years later, things are getting better. In the beginning, I had diarrhea and didn’t feel well.”*
(A06)

They also have to deal with the conflicting preferences of other family members. Since their adult children are also recent immigrants to Shanghai, their daughters-in-law or sons-in-law may come from different places and have different tastes. To cope with this dilemma, the seniors mainly take two strategies. The first is to take their customary way to cook local ingredients. Although the ingredients taste different from those at home due to the climate difference, they still cook them in their familiar way, making more familiar flavors.


*“The wax gourd here tastes different from ours. There is a smell in it. The lotus root is not glutinous either. But we still get used to using them in soups, duck soup and rib soup, which is a typical way of cooking in our hometown.”*
(From the first author’s field notes)

The tastes of all family members are grinding-in in this way, in which the grandparents gradually customize the taste of “home” in this new city and gain a sense of control and self-efficacy. However, such cooking sometimes makes the table monotonous. Hence, a second strategy is to borrow tastes in other ways, such as buying take-out, purchasing their favorite ingredients through mobile apps, and occasionally going to restaurants. Many of the researched grandparents mentioned that they were pretty surprised by the convenience of e-commerce services in Shanghai. These experiences also enhance the seniors’ insight and curiosity to taste modern city life, which benefits their mental wellbeing:


*“We also have express delivery (in our hometown), but it’s not as fast as here……There are packages to pick up almost every day……You can buy anything you want through your phone… I’ve also bought earth eggs a few times……If they (the kids) don’t like the dish of the day, they can also ask for “Wai Mai (take-out food)” and in half an hour, they will arrive. Very convenient.”*
(A16)


*“In Shanghai, I have opened my eyes and learned a lot of fashionable things. Recently, I gradually learned to sort my garbage. I will probably keep this habit when I return to Xuancheng (my home city) in the future.”*
(A29)

### 4.2. Social Practices: Restrictions and Reconstructions

The migrating grandparents’ social wellbeing experience is dynamic and fluctuating, entangling with the social contacts around their two homes. Initially, their social connections were restricted by the physical distance but gradually reconstructed and renewed in real and virtual communities. However, long-term nostalgia for familiar social circles persists in their minds.

Because of the tight schedule, most interviewed migrating grandparents reported their limited social contact with local communities.


*“There are singing and dancing (teams) in the community, but I have no time. Sometimes they (practice) at 8 a.m. or 4 p.m., but I can’t leave. Because there are little kids here, you have to take care of them, right? My wife has to cook. There is no one to pick up my little granddaughter outside, or she must stay at home alone, right? The old ladies are singing and dancing in the community, but I can’t join. No way. If it’s just our old couple here, then it’s possible.”*
(A09)

Hence the migrating grandparents’ social contact within the neighborhoods is quite restricted by the housework. However, simultaneously it is some other housework that pushes them to enter the social circle of the neighborhood community. As many female migrating grandparents have to stroll babies or look after kids in the nearby neighborhood and park, they find a common tribe and start new social contact within it.


*“In Shanghai, there are also friends like us looking after grandchildren. We chat together. When we take our babies out to play, we are mostly the same age. Then we get to know each other.”*
(A06)


*“…We have gathered a group of the same people here. For example, many grandmothers and maternal grandmothers in our neighborhood also come to care for their grandsons or granddaughters. They play in this neighborhood. We are similar. Later, we became familiar with each other. We chat like close friends, talk and play. At leisure, we bring kids to play together, play together there, just like new old friends. Then you will feel a little more delighted, a little happier, and life is not so monotonous, not completely staying home. We feel happy to play with some new friends.”*
(A08)

As A08 described, the tribe taking care of kids enlarges the grandparents’ social contacts and improves their social wellbeing. To some extent, it relieves the loneliness and boredom and attaches freshness and intimacy to their migrating home. Nevertheless, this kind of social contact is shallow and happens more among grandmothers. More often, the internet, smartphone and social media help them open the social world and build connections with their old friends and relatives in their hometown. Just like (C05) said, “In WeChat, we have a group called ‘a loving family’. We send pictures of our daily lives and voices, to chat and talk in it.”

The technologies empower the migrating elders with new platforms for social interaction, especially for those who come from slightly backward rural areas and cannot type words. However, maintaining the social bonds with hometown people may also bring a sense of loss to them:


*“We keep the connection on WeChat...Generally, there are many brothers and sisters of our age. All are retired. Then I’m afraid they ask me, ‘When will you come back? ’Once they ask me, my heart seems to be very uncomfortable. I will tell them that don’t ask me when I will come back. I will tell you when I want to come back. I say that when you ask me, my heart will crack.”*
(A14)

A14’s words reflect these migrating grandparents’ complex feelings towards their mobile life. On the one hand, they insist on staying with their children and supporting their life. On the other hand, relatives and friends in their hometown are long-term acquaintances and represent the belongings and roots of their life. They have to devote themselves to their children’s home, but at the same time, they are pretty missing and reluctant to give up their roots.

### 4.3. Emotional Practices: Attachments and Entanglements

The migrating grandparents devote much energy and emotions to their grandchildren and gradually obtain emotional attachment and spiritual fulfillment in the migrating home. However, emotionally, they are always eager to bridge the distance between the two homes. 

Most interviewed grandmothers identify themselves as babysitters making every effort to take care of the grandchildren:


*“…After arriving here, I stay at home. Say a bad word, I’m a nanny. I’m here to serve them, such as babysitting and doing housework…I want to relieve their burden, not employ a nanny.”*
(A08)


*“Anyway, it’s necessary to do things conscientiously and contribute to our children. (We have) no other requirements. We have shallow requirements for ourselves. We just contribute to others. It’s all true. It is what our generation is. Bearing hardship without complaint, we are just nannies.”*
(A10)

Some grandfathers who can drive also make jokes about themselves being the exclusive family driver in their children’s new home. Although tough and tiring, witnessing the grandchildren’s growth gives them great spiritual satisfaction and enriches their retiring life, improving their mental wellbeing. Just as C05 and B08 commented:


*“……The elderly in China always regards children as the center of our lives. After reaching old age, children are usually not around, and life will feel lost. So, although tired, bringing up the grandchildren feels very fulfilling and meaningful.”*
(C05)


*“They are extremely good with the grandchildren, even closer than their daughters. These days they come back to their hometown and make phone calls (with the grandchildren) every day.”*
(B08)

Hence the intergenerational kinship becomes a vital bond for the grandparents to build a tremendous emotional attachment to the new home, enhancing their internal sense of gain and happiness.

However, at the same time, there is continuously a kind of emotional entanglement between enjoying a family reunion in Shanghai and missing the familiarity and belongings of hometown among the elders. Such tangled emotions change along with their moving between these two places. A21′s words provide a clear description:


*“At first, I was not very satisfied (with life in Shanghai), but now I slowly get used to it, (though) there will always be something unsatisfied. after staying here long, I want to go back. But if I go back and stay at home, I will miss them and want to come back again. Later, my son said to me, ‘You just have a rest in your hometown, and have a reunion when coming to Shanghai.”*
(A21)

Just as A21 describes, although to many migrating grandparents, their material, social and spiritual wellbeing can be satisfied and improved to some extent in the new home, they still feel they are lacking some belongings and familiarity with their hometown. However, they miss the reunion with their children again after returning to their hometown. 

Some migrating grandparents use the “on duty by turns” strategy to balance such emotional entanglement. For example, A21 and A22 are relatives by marriage. They take turns visiting Shanghai and caring for the same young couple. They initiatively increase the rotating frequency between Shanghai and their hometown and become highly mobile migrating “birds”. In the subconsciousness, they hope to balance their attachments to the two homes and piece together a complete home. Such high mobility eased their emotional tension in the two homes and enriched their holistic wellbeing. Just as C03 said, 


*“We just pretend to be travelling each time we switch the place… to feel the different lives in both the big and small cities is also a wonderful thing”.*


However, this high-frequent mobile lifestyle is not common among migrating grandparents. After all, it requires close cooperation among the family members, good economic situations, and a strong body.

## 5. Discussion and Conclusions

### 5.1. Research Summary

How to improve the wellbeing of the elderly is a worldwide issue. By absorbing the multi-disciplinary thoughts of wellbeing, health geographers call for a relational and processual approach to understanding elders’ wellbeing experiences in dynamic human–place relationships. From the relational perspective, the home is de-territorialized without being fixed to a specific location. Rather than bounded by physical distance, it is separated by social–relational distance crossing multi-scales. Hence, it provides a new perspective to dig into the elder–home relations and their wellbeing experience in a moving lifestyle. This study concentrates on a unique group of migrating grandparents who rotate domestically between their own homes and their adult children’s homes to support the children’s living. It provides a particular case to explore home-based wellbeing in a rotating lifestyle.

Through analyzing the interviews and the first author’s field notes, we find that the migrating grandparents’ wellbeing experiences are dynamically negotiated in their material, social and emotional home-making practices. Through their corporeal and social body, the elders experience the differences between the two homes in various material aspects such as housework, use of home space, household expenses and diets. It significantly interrupts their life pace and challenges the physical strength and adaptability of their body. They take initiatives such as squeezing time to rest, making space for their offspring, cooking in the familiar style or ordering some delicious food to overcome the physical and mental tiredness and discomfort. They also receive more economic and medical support from their offspring, which can be seen as additional safeguards for their physical and mental wellbeing. In the social aspect, the grandparents have to endure the physical separation from their familiar social circle in their homeland, although some can build external social connections with the neighbors in the migrating home. Communicative technologies such as WeChat rebuild their links with the hometown people, thus relieving their loneliness and improving their social wellbeing. However, long-term homesickness persists in their imagination of the original home. In the emotional aspect, supporting their adult children’s careers and accompanying grandchildren’s growth greatly adds value to the retirement stage of these grandparents, improving their mental wellbeing. 

However, in Chinese culture, the co-existence of “Jia” and “Hu” in the same space and time consist of a completed home [75]. “Jia” means the social unit connected by lineal kin family members and “Hu” means the familiar way of living. However, to these migrating grandparents, they have to spiritually endure and bridge the tension that “Jia” and “Hu” are separated. Hence at any time, these migrating grandparents cannot achieve an optimal sense of wellbeing, but have to make compromises, adjustments and innovations in various home-making practices, thus negotiating a suboptimal sense of wellbeing in balancing the physical, social and mental dimensions.

### 5.2. Theoretical Contributions

This study makes theoretical contributions to the intersection of elderly wellbeing and home-making studies by incorporating the relational approach. In the study of elderly wellbeing, previous studies have revealed the elders’ home-based wellbeing experiences from material, social–cultural, emotional and mental dimensions [4,29,71,72]. Studies about the amenity-led seasonal retirees put forward the complexity of the two homes attached to the migrating seniors in influencing their wellbeing experience [17,64]. Based on them, this study presents a different pattern of how migrating elders negotiate their wellbeing related to two separated homes. Instead of consumption-led migration for better living conditions, they move to support their children, which means they must devote hard work to the migrating home. Hence, unlike the seasonal retirees who seek and enjoy the amenities of a destination city, the migrating grandparents mainly devote their bodies to the children’s home and have less time to enjoy the attractions of a destination city. It relatively weakens their physical wellbeing. Concerning the living conditions, the seasonal retirees always live in the second home independently. However, the migrating grandparents mostly have to share the home space with their children, thus considering other family members in every aspect of the home-making practices. This affects their emotional attachment with the downward generations, improving their long-term mental and spiritual wellbeing compared to the seasonal retirees. 

In the home-making literature, previous studies connecting migrating elders and home-making practices have presented their roles, responsibilities, and relationships with the host destination [20,21,50,51,52]. The relational approach provides a broader perspective to reconsider the elders’ dynamic and complex relationship to both the original and the migrating homes in their daily home-making practices. Compared to the transnational migrating grandparents, the domestic Chinese grandparents have more flexibility and negotiability in the home-making practices. They can rotate between the children’s and their own homes more frequently and flexibly. In the children’s home, they receive more material and spiritual fulfillment and gradually adapt to the pace of metropolitan life. In their hometown, they can recover from physical tiredness, regain a free and relaxed body, and have a familiar social community. The rotating lifestyle makes it possible to relieve the tensions between the physical, social and mental wellbeing and achieve a suboptimal balance in dynamics.

Moreover, previous studies show that western grandparents primarily provide their adult children with short-term support only at some important life moments, but Asian grandparents can stay longer and persist in this two-home rotating lifestyle [21,53]. This study supplements this conversation by highlighting the contemporary Chinese context. Chinese urbanization attracts a large population to move into cities and take root, forcing the separation between “Jia” and “Hu”. To survive and win the high-quality medical, occupational and educational resources in big cities, the small family of three generations united as an independent social unit through stable and fixed lineal blood kinship [76]. These provide spiritual energies for the migrating grandparents to overcome difficulties and willingly make sacrifices during mobile life. This, to a certain extent, improves Chinese elders’ mental wellbeing and contributes to their long-term persistence. 

### 5.3. Practical Implications

This study has some practical implications. As the separation between the lineal kinship and the homeland forms the central tension for improving the migrating grandparents’ wellbeing experience, some supportive work could be carried out in the migrating cities. Specific measures include building communicative groups such as WeChat chatting groups for fellow-hometown migrating elders, constructing more amenity spaces such as gardens and parks around communities for the migrating grandparents to conduct grandchildren care, socialization and recreation, and providing consultation and guidance to the migrating family members. These external supports are also necessary to relieve the tension and help the grandparents achieve suboptimal wellbeing. 

### 5.4. Limitations and Future Studies

This study has some limitations. Because of the refusal of informed consent, the sample size of the formal interview is limited. The triangulation could be partly limited by interviewing only nine adult children of the 35 interviewed grandparents. We make up for this defect by collecting field notes through informal talks with the migrating grandparents. Future studies could combine other qualitative methods such as observations and auto-ethnography to collect more detailed data about the migrating grandparents’ wellbeing experience along with the change in time, places and socio-demographic characteristics. For example, this study indicates that gender, age, previous working experiences and the migrating time are important factors influencing the grandparents’ home-making practices, through which different place-based wellbeing experiences would be produced. 

## Figures and Tables

**Figure 1 ijerph-19-09903-f001:**
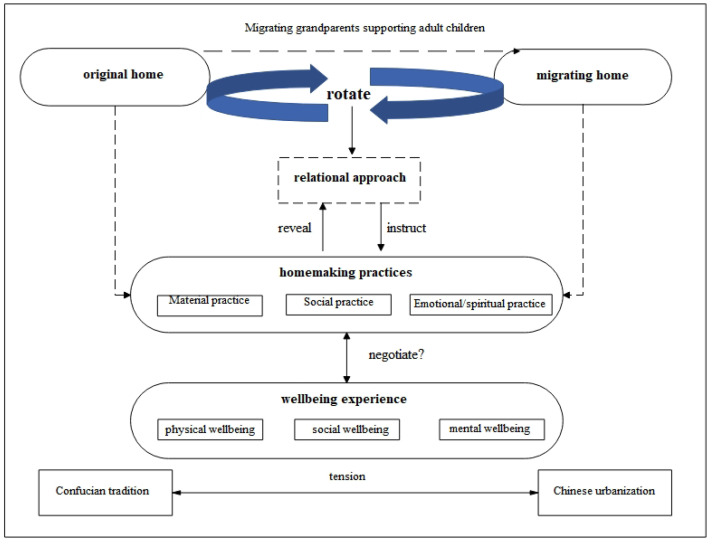
The research framework.

**Table 1 ijerph-19-09903-t001:** Conventional and relational perspectives of place and place-based wellbeing.

Approaches	Understanding of Place	Understanding of Wellbeing	References
Conventional approach	1. Space at a specific scale, separated by physical distance 2. Local residents living in a particular place 3. Fixed geographical boundaries 4. Context keeps consistent with different individuals and groups	1. Individually distinguishing the contextual (physical and social contexts) and compositional (e.g., the human-related characteristics) effects of place on peoples’ wellbeing 2. Wellbeing as measurable through identification of components	[2,4,8,30,31]
Relational approach	1. Fluid networks, multi-scale, separated by social–relational distance 2. People are mobile in diverse temporal–spatial patterns 3. Dynamic territorial divisions changed according to social relations, cultures, discourses, etc. 4. Context is variable with different individuals and groups	1. Exploring human–place interactions over time in which wellbeing is negotiated in this process 2. People are constantly negotiating among the dynamic physical, social and mental dimensions of wellbeing experiences	[2,4,5,7,27,28,29,32]

**Table 2 ijerph-19-09903-t002:** Profile of the interviewed migrating grandparents.

No.	Gender	Age	Hometown	Moving Time	Back to Hometown Frequency
Respondent Information in Stage 1					
A01	Female	60	Hubei	10 years	Returns in case of problems
A02	Female	59	Huzhou, Zhejiang	4 years	Returns for festivals
A03	Female	70	Jiangsu	4 years	Returns in case of problems
A04	Female	56	Dalian, Liaoning	Unknown	Returns in case of problems
A05	Female	60	Xi’an, Shaanxi	6 years	Problems and festivals
A06	Female	60	Jiangxi	5.5 years	Returns for festivals
A07	Female	55	Shandong	3 years	Returns on National Day, Spring Festival, Father’s birthday
A08	Female	68	Zhenjiang, Jiangsu	13 years	Returns for festivals
A09	Female	70	Nanchang, Jiangxi	8 years	Returns in case of problems
A10	Female	69	Jingjiang, Jiangsu	21 years	Once to twice a year
A11	Male	72	Anhui	5.5 years	1–2 months each year
A12	Female	66	Lianyungang, Jiangsu	5.5 years	Once a year
A13	Female	66	Shangrao, Jiangxi	15 years	Once a year
A14	Male	59	Fuzhou, Fujian	5 years	Returns in winter and summer vacation
A15	Female	57	Songyuan, Jilin	4 years	4–8 months each year
A16	Female	73	Yangzhou, Jiangsu	13 years	Twice a year (Spring Festival, National Day)
A17	Male	68	Chizhou, Anhui	21 years	About 3 times a year (busy farming season)
A18	Female	69	Zhenjiang, Jiangsu	3 years	About 6 times a year
A19	Female	65	Chengdu, Sichuan	7 years	Twice a year
A20	Female	71	Dangshan, Anhui	10 years	About 3 times a year (busy farming season)
A21	Female	63	Nantong, Jiangsu	5 years	About 3 times a year, rotate with A22
A22	Male	63	Hefei, Anhui	5 years	About 3 times a year, rotate with A21
A23	Female	58	Nantong, Jiangsu	9 years	About three times a year (busy farming season)
A24	Male	75	Yichun, Jiangxi	12 years	3 times a year (Qing Ming and Spring Festival, National Day)
A25	Female	58	Huaian, Jiangsu	3 years	Twice a year
A26	Female	51	Hefei, Anhui	3 years	About 10 times a year (personal affairs)
A27	Female	63	Suzhou, Jiangsu	2 years	4 times a year (personal affairs and rest)
A28	Female	53	Jingdezhen, Jiangxi	3 years	Twice a year
A29	Male	68	Xuancheng, Anhui	3 years	About 7 times a year
A30	Female	69	Xuzhou, Jiangsu	2 years	3 times a year
Respondent Information in Stage 2					
C01	Female	56	Wuhan, Hubei	6 years	2–3 times a year
C02	Male	62	Wuhan, Hubei	6 years	2–3 times a year
C03	Female	57	Huangshan, Anhui	about 4 years	2–3 times a year
C04	Male	64	Huangshan, Anhui	about 4 years	2–3 times a year
C05	Female	62	Hangzhou, Zhejiang	Several years (not specific)	About 3 times a year

**Table 3 ijerph-19-09903-t003:** Profile of the interviewed adult children (conducted in Stage 1).

No.	Gender	Work in Shanghai	Family Relationship
B01	Female	Company treasurer	Daughter-in-law of A15
B02	Female	Florist	Daughter-in-law of A16
B03	Female	Civil servant	Daughter of A18
B04	Male	Unknown	Son of A20
B05	Female	Civil servant	Daughter of A21, daughter-in-law of A22
B06	Female	Civil servant	Daughter of A23
B07	Male	Civil servant	Son of A25
B08	Female	Graduate student	Daughter of A26
B09	Female	Civil servant	Daughter of A27

**Table 4 ijerph-19-09903-t004:** Definitions and exemplary labels of the categories/subcategories.

Concept	Categories	Explanations	Exemplary Labels	References
Home-making practice	Material home-making practices	Material practices to maintain physiological and psychological security and stability	Housework, food and expenditure arrangement, usage of the material space, facilities, goods in relation to two homes	[17,20,25,42,47,64,69,70]
	Social home-making practices	Social practices building and maintaining familiar social network	Socialization with familiar friends and relatives, making social bonds at multi-scales	
	Emotional/spiritual home-making practices	Practices to generate emotional or spiritual attachments	Communicate and build emotional bonds with the family members, communities, migrating and homeland places	
Wellbeing experience	Physical wellbeing	The physiological/physical health of human body	Physical health, physical strength; easy body	[4,29,71,72]
	Social wellbeing	A good state of social contacts and supports	Fulfillment of social needs, feel socially connected with friends and relatives	
	Mental wellbeing	Peoples’ positive feelings, emotions and spiritual attachments	Spiritual fulfillment, mental happiness and enjoyment; satisfaction with life in the psychological sense	

## Data Availability

All related data and methods are presented in this paper and the Table 2 and Table 3. Additional inquiries should be addressed to the corresponding author.

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
