# Peer review of "The Wellbeing of Chinese Migrating Grandparents Supporting Adult Children: Negotiating in Home-Making Practices"

_ijerph, 2022, doi:10.3390/ijerph19169903_

Round 1

Reviewer 2 Report

I believe the research presented lacks theoretical focus and an appropriate methodology. A qualitative study requires more empirical work in terms of operationalizing the concepts the authors use and perhaps proving a descriptive table of the various social interactions observed.

I believe this paper is more about the 'negotiated' social interactions that occur when grandparents travel to live with their children. Some of your observations are confusing. For example, you write: "The seniors who are busy moving between their homes and their children’s homes to support their children’s family are not carefully examined. Rotating between the two homes becomes a lifestyle for these grandparents. Sometimes, they leave their hometown and live with their children and grandchildren for a long time to undertake the home-caring responsibilities. Intermittently they go back to their own home and live their own life."

I suspect you need to focus on one type (group) of migrating grandparents and observe their particular behaviors. It is quite possible that you have 3 different papers you could do from your main focus.

The following is an especially confusing sentence (and too long): "through re-connecting with children and relatives in hometowns, building a familiar social network and political participation in the destination, they rebuild the social home; and through sustaining routines between these two homes, they reflect their feelings of estrangement in destination society and desire to return to the original home, thus to renegotiate their personal home.

I'm not sure what you are revealing here. You use terms like 'dynamic' and 'relational', but which phenomenon is dynamic and which is relational. Moreover, can you construct those and then empirically observe?

In your conclusion you write: "Hence, their adult children's care gives these grandparents some psychological comfort. All the interviewed members expressed that they would insist on supporting their children as long as their body permits." 

So what does this mean? In a Confucian society, isn't this the norm? I really believe you should consider observing normative and deviations to better describe seniors moving to live with children.

Round 2

Reviewer 1 Report

Authors did a great work and addressed the suggested revisions very well. The paper is currently very clear, and with useful integrations and explanations. Also, Authors carefully explained how they addressed  the revisions themselves.

Reviewer 2 Report

The resubmission is more to the point of the research and makes a contribution to the symbolic interactionist literature on grandparents, their adult children, and their new, mixed family living arrangement. Well done!

This manuscript is a resubmission of an earlier submission. The following is a list of the peer review reports and author responses from that submission.